# Beta Embeddings for Multi-Hop Logical Reasoning in Knowledge Graphs

**Hongyu Ren**
Stanford University
hyren@cs.stanford.edu

**Jure Leskovec**
Stanford University
jure@cs.stanford.edu

## Abstract

One of the fundamental problems in Artificial Intelligence is to perform complex multi-hop logical reasoning over the facts captured by a knowledge graph (KG). This problem is challenging, because KGs can be massive and incomplete. Recent approaches embed KG entities in a low dimensional space and then use these embeddings to find the answer entities. However, it has been an outstanding challenge of how to handle arbitrary first-order logic (FOL) queries as present methods are limited to only a subset of FOL operators. In particular, the negation operator is not supported. An additional limitation of present methods is also that they cannot naturally model uncertainty. Here, we present BETAE, a probabilistic embedding framework for answering arbitrary FOL queries over KGs. BETAE is the first method that can handle a complete set of first-order logical operations: conjunction ($\wedge$), disjunction ($\vee$), and negation ($\neg$). A key insight of BETAE is to use probabilistic distributions with bounded support, specifically the Beta distribution, and embed queries/entities as distributions, which as a consequence allows us to also faithfully model uncertainty. Logical operations are performed in the embedding space by neural operators over the probabilistic embeddings. We demonstrate the performance of BETAE on answering arbitrary FOL queries on three large, incomplete KGs. While being more general, BETAE also increases relative performance by up to 25.4% over the current state-of-the-art KG reasoning methods that can only handle conjunctive queries without negation.

## 1 Introduction

Reasoning is a process of deriving logical conclusion or making predictions from available knowledge/facts. Knowledge can be encoded in a knowledge graph (KG), where entities are expressed as nodes and relations as edges. Real-world KGs, such as Freebase [1], Yago [2], NELL [3], are large-scale as well as noisy and incomplete. Reasoning in KGs is a fundamental problem in Artificial Intelligence. In essence, it involves answering first-order logic (FOL) queries over KGs using operators existential quantification ($\exists$), conjunction ($\wedge$), disjunction ($\vee$), and negation ($\neg$).

To find answers, a given FOL query can be viewed as a computation graph which specifies the steps needed. A concrete example of the computation graph for the query "*List the presidents of European countries that have never held the World Cup*" is shown in Fig. 1. The query can be represented as a conjunction of three terms: "*Located(Europe,V)*", which finds all European countries; "*¬Held(World Cup,V)*", which finds all countries that never held the World Cup; and "*President(V,V?)*", which finds presidents of given countries. In order to answer this query, one first locates the entity "*Europe*" and then traverses the KG by relation "*Located*" to identify a set of European countries. Similar operations are needed for the entity "*World Cup*" to obtain countries that hosted the World Cup. One then needs to complement the second set to identify countries that have never held the World Cup and intersect the complement with the set of European countries. The final step is to apply the relation

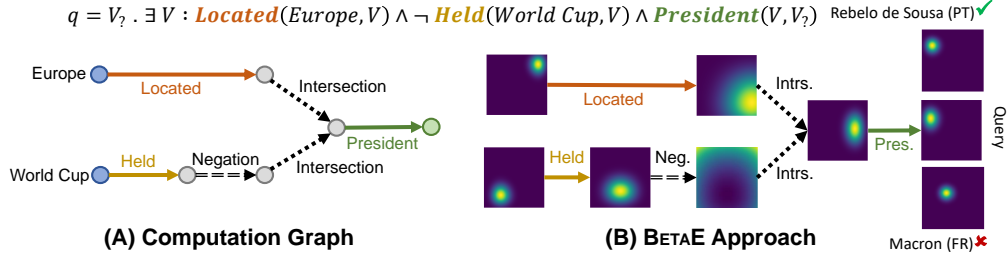

Figure 1: BETAE **answers first-order logic queries that include** $\exists$**,** $\wedge$**,** $\vee$ **and** $\neg$ **logical operators.**
(A): A given query *"List the presidents of European countries that have never held the World Cup"* can be represented by its computation graph where each node represents a set of entities and each edge represents a logical operation. (B): BETAE models each node of the computation graph as a Beta distribution over the entity embedding space and each edge of the computation graph transforms the distribution via a projection, negation, or intersection operation. BETAE applies a series of logical operators that each transform and shape the Beta distribution. The answer to the query are then entities that are probabilistically close to the embedding of the query (*e.g.*, embedding of "Macron" is closer to the query embedding and the embedding of "Rebelo de Sousa").

"*President*" to the resulting intersection set to find the list of country presidents, which gives the query answer.

KG reasoning presents a number of challenges. One challenge is the scale of KGs. Although queries could be in principle answered by directly traversing the KG, this is problematic in practice since multi-hop reasoning involves an exponential growth in computational time/space. Another challenge is incompleteness, where some edges between entities are missing. Most real-world KGs are incomplete and even a single missing edge may make the query unanswerable.

Previous methods [4, 5, 6, 7, 8] aim to address the above challenges by using embeddings and this way implicitly impute the missing edges. Methods also embed logical queries into various geometric shapes in the vector space [9, 10, 11, 12]. The idea here is to design neural logical operators and embed queries iteratively by executing logical operations according to the query computation graph (Fig. 1). An advantage of these approaches is that they do not need to track all the intermediate entities, and that they can use the nearest neighbor search [13] in the embedding space to quickly discover answers. However, these methods only support existential positive first-order (EPFO) queries, a subset of FOL queries with existential quantification ($\exists$), conjunction ($\wedge$) and disjunction ($\vee$), but not negation ($\neg$). Negation, however, is a fundamental operation and required for the complete set of FOL operators. Modeling negation so far has been a major challenge. The reason is that these methods embed queries as closed regions, *e.g.*, a point [9, 11, 12] or a box [10] in the Euclidean space, but the complement (negation) of a closed region does not result in a closed region. Furthermore, current methods embed queries as static geometric shapes and are thus unable to faithfully model uncertainty.

Here we propose *Beta Embedding (*BETAE*)*, a method for multi-hop reasoning over KGs using full first-order logic (FOL). We model both the entities and queries by probabilistic distributions with bounded support. Specifically, we embed entities and queries as Beta distributions defined on the $[0, 1]$ interval. Our approach has the following important advantages: (1) Probabilistic modeling can effectively capture the uncertainty of the queries. BETAE adaptively learns the parameters of the distributions so that the uncertainty of a given query correlates well with the differential entropy of the probabilistic embedding. (2) We design neural logical operators that operate over these Beta distributions and support full first-order logic: $\exists$, $\wedge$, $\vee$ and most importantly $\neg$. The intuition behind negation is that we can transform the parameters of the Beta distribution so that the regions of high probability density become regions of low probability density and vice versa. (3) Our neural modeling of $\wedge$ and $\neg$ naturally corresponds to the real operations and captures several properties of first-order logic. For example, applying the negation operator twice will return the same input. (4) Using the De Morgan's laws, disjunction $\vee$ can be approximated with $\wedge$ and $\neg$, allowing BETAE to handle a complete set of FOL operators and thus supporting arbitrary FOL queries.

Our model is able to handle arbitrary first-order logic queries in an efficient and scalable manner. We perform experiments on standard KG datasets and compare BETAE to prior approaches [9, 10] that can only handle EPFO queries. Experiments show that our model BETAE is able to achieve state-of-the-art performance in handling arbitrary conjunctive queries (including $\exists$, $\wedge$) with a relative

increase of the accuracy by up to 25.4%. Furthermore, we also demonstrate that BETAE is more general and is able to accurately answer any FOL query that includes negation $\neg$. Project website with data and code can be found at `http://snap.stanford.edu/betae`.

## 2 Related Work

**Uncertainty in KG Embeddings**. Previous works on KG embeddings assign a learnable vector for each entity and relation with various geometric intuitions [4, 5, 6, 7, 8] and neural architectures [14, 15, 16]. Besides vector embeddings, KG2E [17] and TransG [18] both model the uncertainties of the entities and relations on KGs by using the Gaussian distributions and mixture models. However, their focus is link prediction and it is unclear how to generalize these approaches to multi-hop reasoning with logical operators. In contrast, our model aims at multi-hop reasoning and thus learns probabilistic embeddings for complex queries and also designs a set of neural logical operators over the probabilistic embeddings. Another line of work models the uncertainty using order embeddings [19, 20, 21, 22, 23], distributions [17, 24, 25] and Quantum logic [26]. The difference here is that our goal is to model the logical queries and their answers, which goes beyond modeling the inclusion and entailment between a pair of concepts in KGs.

**Multi-hop Reasoning on KGs**. Another line of related work is multi-hop reasoning on KGs. This includes (1) answering multi-hop logical queries on KGs, which is most relevant to our paper, and (2) using multi-hop rules or paths to improve the performance of link prediction. Previous methods that answer queries [9, 10, 11, 12] can only model a subset of FOL queries, while our method can handle arbitrary FOL queries with probabilistic embeddings. Rule and path-based methods [27, 28, 29, 30, 31, 32] pre-define or achieve these multi-hop rules in an online fashion that require a modeling of all the intermediate entities on the path, while our main focus is to directly embed and answer a complex FOL query without the need to model the intermediate entities, which leads to more scalable algorithms.

## 3 Preliminaries

Knowledge Graph (KG) $\mathcal{G}$ is heterogeneous graph structure that consists of a set of entities $\mathcal{V}$ and a set of relation types $\mathcal{R}$, $\mathcal{G} = (\mathcal{V}, \mathcal{R})$. Each relation type $r \in \mathcal{R}$ is a binary function $r : \mathcal{V} \times \mathcal{V} \to \{\texttt{True}, \texttt{False}\}$ that indicates (directed) edges of relation type $r$ between pairs of entities.

We are interested in answering first-order logic (FOL) queries with logical operations including conjunction ($\wedge$), disjunction ($\vee$), existential quantification ($\exists$) and negation ($\neg$) [1]. We define valid FOL queries in its disjunctive normal form (DNF), *i.e.*, disjunction of conjunctions.

**Definition 1** (First-order logic queries)**.** *A first-order logic query $q$ consists of a non-variable anchor entity set $\mathcal{V}_a \subseteq \mathcal{V}$, existentially quantified bound variables $V_1, \ldots, V_k$ and a single target variable $V_?$, which provides the query answer. The disjunctive normal form of a logical query $q$ is a disjunction of one or more conjunctions.*

$$q[V_?] = V_? \, . \, \exists V_1, \ldots, V_k : c_1 \vee c_2 \vee \ldots \vee c_n$$

1. *Each $c$ represents a conjunctive query with one or more literals $e$. $c_i = e_{i1} \wedge e_{i2} \wedge \cdots \wedge e_{im}$.*

2. *Each literal $e$ represents an atomic formula or its negation. $e_{ij} = r(v_a, V)$ or $\neg \, r(v_a, V)$ or $r(V', V)$ or $\neg \, r(V', V)$, where $v_a \in \mathcal{V}_a$, $V \in \{V_?, V_1, \ldots, V_k\}$, $V' \in \{V_1, \ldots, V_k\}$, $V \neq V'$, $r \in \mathcal{R}$.*

**Computation Graph:** As shown in Fig. 1, we can derive, for a given query, its corresponding computation graph by representing each atomic formula with relation projection, merging by intersection and transforming negation by complement. This directed graph demonstrates the computation process to answer the query. Each node of the computation graph represents a distribution over a set of entities in the KG and each edge represents a logical transformation of this distribution. The computation graphs of FOL queries can be viewed as heterogeneous *trees*, where each leaf node corresponds to a set of cardinality 1 that contains a single anchor entity $v_a \in \mathcal{V}_a$ (note that one anchor entity may

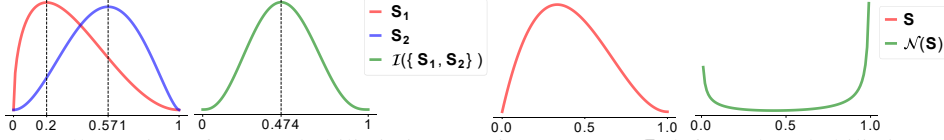

Figure 2: Illustration of our probabilistic intersection operator $\mathcal{I}$ (left) and probabilistic negation operator $\mathcal{N}$ (right). $\mathcal{I}$ transforms the input distribution by taking the weighted product of the PDFs; $\mathcal{N}$ transforms the input distribution by taking the reciprocal of its parameters.

appear in multiple leaf nodes) and the root node represents the unique target variable, which is the set of answer entities. The mapping along each edge applies a certain logical operator:

1. **Relation Projection:** Given a set of entities $S \subseteq \mathcal{V}$ and relation type $r \in \mathcal{R}$, compute adjacent entities $\cup_{v \in S} A_r(v)$ related to $S$ via $r$: $A_r(v) \equiv \{v' \in \mathcal{V} : r(v, v') = \texttt{True}\}$.

2. **Intersection:** Given sets of entities $\{S_1, S_2, \ldots, S_n\}$, compute their intersection $\cap_{i=1}^{n} S_i$.

3. **Complement/Negation:** Given a set of entities $S \subseteq \mathcal{V}$, compute its complement $\overline{S} \equiv \mathcal{V} \setminus S$.

We do not define a union operator for the computation graph, which corresponds to disjunction. However, this operator is not needed, since according to the De Morgan's laws, given sets of entities $\{S_1, \ldots, S_n\}$, $\cup_{i=1}^{n} S_i$ is equivalent to $\overline{\cap_{i=1}^{n} \overline{S}}$.

In order to answer a given FOL query, we can follow the computation graph and execute logical operators. We can obtain the answers by looking at the entities in the root node. We denote the answer set as $[\![q]\!]$, which represents the set of entities on $\mathcal{G}$ that satisfy $q$, *i.e.*, $v \in [\![q]\!] \iff q[v] = \texttt{True}$. Note that this symbolic traversal of the computation graph is equivalent to traversing the KG, however, it cannot handle noisy or missing edges in the KG.

## 4 Probabilistic Embeddings for Logical Reasoning

To answer queries in a large and incomplete KG, we first introduce our model BETAE, which embeds both entities and queries as Beta distributions. Then we define probabilistic logical operators for relation projection, intersection and negation. These operate on the Beta embeddings which allow us to support arbitrary FOL queries. Finally, we describe our training objective.

### 4.1 Beta Embeddings for Entities and Queries

In order to model any FOL query, the desirable properties of the embedding include: (1) the embedding can naturally model uncertainty; (2) we can design logical/set operators (conjunction/intersection and especially negation/complement) that are closed. The closure property is important for two reasons: (i) operators can be combined in arbitrary ways; (ii) the representation remains at a fixed space/time complexity and does not grow exponentially as additional operators are applied.

We propose to embed both the entities and queries into the same space using probabilistic embeddings with bounded support. With a bounded support, the negation/complement can be accordingly defined, where we follow the intuition to switch high-density regions to low density and vice versa (Fig. 2). Specifically, we look at the $[0, 1]$ interval and adopt the Beta distribution. A Beta distribution $\texttt{Beta}(\alpha, \beta)$ has two shape parameters, and our method relies on its probability density function (PDF): $p(x) = \frac{x^{\alpha-1}(1-x)^{\beta-1}}{\mathbf{B}(\alpha, \beta)}$, where $x \in [0, 1]$ and $\mathbf{B}(\cdot)$ denotes the beta function. The uncertainty of a Beta distribution can be measured by its differential entropy: $H = \ln \mathbf{B}(\alpha, \beta) - (\alpha - 1)[\psi(\alpha) - \psi(\alpha + \beta)] - (\beta - 1)[\psi(\beta) - \psi(\alpha + \beta)]$, where $\psi(\cdot)$ represents the digamma function.

For each entity $v \in \mathcal{V}$, which can be viewed as a set with a single element, we assign an initial Beta embedding with learnable parameters. We also embed each query $q$ with a Beta embedding, which is calculated by a set of probabilistic logical operators (introduced in the next section) following the computation graph. Note that BETAE learns high-dimensional embeddings where each embedding consists of multiple independent Beta distributions, capturing a different aspect of a given entity or a query: $\mathbf{S} = [(\alpha_1, \beta_1), \ldots, (\alpha_n, \beta_n)]$, where $n$ is a hyperparameter. We denote the PDF of the $i$-th Beta distribution in $\mathbf{S}$ as $p_{\mathbf{S}, \mathbf{i}}$. Without loss of generality and to ease explanation, we shall assume that each embedding only contains one Beta distribution: $\mathbf{S} = [(\alpha, \beta)]$, and we denote its PDF as $p_{\mathbf{S}}$.

## 4.2 Probabilistic Logical Operators

In order to answer a query using the computation graph, we need probabilistic logical operators for the Beta embedding. Next, we describe the design of these logical operators used in computation graphs, which include relation projection $\mathcal{P}$, intersection $\mathcal{I}$ and negation $\mathcal{N}$. As discussed before, union can be implemented using intersection and complement. Each operator takes one or more Beta embeddings as input and then transforms them into a new Beta embedding.

**Probabilistic Projection Operator $\mathcal{P}$:** In order to model the relation projection from one distribution to another, we design a probabilistic projection operator $\mathcal{P}$ that maps from one Beta embedding $\mathbf{S}$ to another Beta embedding $\mathbf{S}'$ given the relation type $r$. We then learn a transformation neural network for each relation type $r$, which we implement as a multi-layer perceptron (MLP):

$$\mathbf{S}' = \mathtt{MLP}_r(\mathbf{S}) \tag{1}$$

The goal here is that for all entities $S$ covered by the input distribution, we can achieve the embedding distribution that covers entities $S' = \cup_{v \in S} A_r(v)$, where $A_r(v) \equiv \{v' \in \mathcal{V} : r(v, v') = \mathtt{True}\}$. Importantly, projection operation represents a relation traversal from one (fuzzy) set of entities to another (fuzzy) set, and may yield a huge number of results, yet here we represent it with a single fixed-size Beta embedding, making BETAE scalable.

**Probabilistic Intersection Operator $\mathcal{I}$:** Given $n$ input embeddings $\{\mathbf{S_1}, \ldots, \mathbf{S_n}\}$, the goal of probabilistic intersection operator $\mathcal{I}$ is to calculate the Beta embedding $\mathbf{S_{Inter}}$ that represents the intersection of the distributions (*i.e.*, the intersection of the distributions defining fuzzy input sets of entities). We model $\mathcal{I}$ by taking the weighted product of the PDFs of the input Beta embeddings:

$$p_{\mathbf{S_{Inter}}} = \frac{1}{Z} \prod p_{\mathbf{S_1}}^{w_1} \ldots p_{\mathbf{S_n}}^{w_n}, \tag{2}$$

where $Z$ is a normalization constant and $w_1, \ldots, w_n$ are the weights with their sum equal to 1.

To make the model more expressive, we use the attention mechanism and learn $w_1, \ldots, w_n$ through a $\mathtt{MLP_{Att}}$ that takes as input the parameters of $\mathbf{S_i}$ and outputs a single attention scalar:

$$w_i = \frac{\exp(\mathtt{MLP_{Att}}(\mathbf{S_i}))}{\sum_j \exp(\mathtt{MLP_{Att}}(\mathbf{S_j}))} \tag{3}$$

Since $\mathbf{S_i}$ is a Beta distribution $[(\alpha_i, \beta_i)]$, the weighted product $p_{\mathbf{S_{Inter}}}$ is a linear interpolation of the parameters of the inputs. We derive the parameters of $\mathbf{S_{Inter}}$ to be $[(\sum w_i \alpha_i, \sum w_i \beta_i)]$:

$$p_{\mathbf{S_{Inter}}}(x) \propto x^{\sum w_i(\alpha_i - 1)} (1 - x)^{\sum w_i(\beta_i - 1)}$$
$$= x^{\sum w_i \alpha_i - 1} (1 - x)^{\sum w_i \beta_i - 1} \tag{4}$$

Our approach has three important advantages (Fig. 2): (1) Taking a weighted product of the PDFs demonstrates a zero-forcing behavior [33] where the effective support of the resulting Beta embedding $\mathbf{S_{Inter}}$ approximates the intersection of the effective support of the input embeddings (effective support meaning the area with sufficiently large probability density [33]). This follows the intuition that regions of high density in $p_{\mathbf{S_{Inter}}}$ should have high density in the PDF of all input embeddings $\{p_{\mathbf{S_1}}, \ldots, p_{\mathbf{S_n}}\}$. (2) As shown in Eq. 4, the probabilistic intersection operator $\mathcal{I}$ is closed, since the weighted product of PDFs of Beta distributions is proportional to a Beta distribution. (3) The probabilistic intersection operator $\mathcal{I}$ is commutative w.r.t the input Beta embeddings following Eq. 2.

**Probabilistic Negation Operator $\mathcal{N}$:** We require a probabilistic negation operator $\mathcal{N}$ that takes Beta embedding $\mathbf{S}$ as input and produces an embedding of the complement $\mathcal{N}(\mathbf{S})$ as a result. A desired property of $\mathcal{N}$ is that the density function should reverse in the sense that regions of high density in $p_{\mathbf{S}}$ should have low probability density in $p_{\mathcal{N}(\mathbf{S})}$ and vice versa (Fig. 2). For the Beta embeddings, this property can be achieved by taking the reciprocal of the shape parameters $\alpha$ and $\beta$: $\mathcal{N}([(\alpha, \beta)]) = [(\frac{1}{\alpha}, \frac{1}{\beta})]$. As shown in Fig. 2, the embeddings switch from bell-shaped unimodal density function with $1 < \alpha, \beta$ to bimodal density function with $0 < \alpha, \beta < 1$.

**Proposition 1.** *By defining the probabilistic logical operators $\mathcal{I}$ and $\mathcal{N}$, BETAE has the following properties (with proof in Appendix A):*

    *1. Given Beta embedding $\mathbf{S}$, $\mathbf{S}$ is a fixed point of $\mathcal{N} \circ \mathcal{N}$: $\mathcal{N}(\mathcal{N}(\mathbf{S})) = \mathbf{S}$.*

2. *Given Beta embedding* $\mathbf{S}$*, we have* $\mathcal{I}(\{\mathbf{S}, \mathbf{S}, \ldots, \mathbf{S}\}) = \mathbf{S}$.

Proposition 1 shows that our design of the probabilistic intersection operator and the probabilistic negation operator achieves two important properties that obey the rules of real logical operations.

### 4.3 Learning Beta Embeddings

**Distance:** Assume we use a $n$-dimensional Beta embedding for entities and queries, which means that each embedding consists of $n$ independent Beta distributions with $2n$ number of parameters. Given an entity embedding $\mathbf{v}$ with parameters $[(\alpha_1^v, \beta_1^v), \ldots, (\alpha_n^v, \beta_n^v)]$, and a query embedding $\mathbf{q}$ with parameters $[(\alpha_1^q, \beta_1^q), \ldots, (\alpha_n^q, \beta_n^q)]$, we define the distance between this entity $v$ and the query $q$ as the sum of KL divergence between the two Beta embeddings along each dimension:

$$\texttt{Dist}(v; q) = \sum_{i=1}^{n} \texttt{KL}(p_{\mathbf{v,i}}; p_{\mathbf{q,i}}), \tag{5}$$

where $p_{\mathbf{v,i}}$ ($p_{\mathbf{q,i}}$) represents the $i$-th Beta distribution with parameters $\alpha_i^v$ and $\beta_i^v$ ($\alpha_i^q$ and $\beta_i^q$). Note that we use $\texttt{KL}(p_{\mathbf{v,i}}; p_{\mathbf{q,i}})$ rather than $\texttt{KL}(p_{\mathbf{q,i}}; p_{\mathbf{v,i}})$ so that the query embeddings will "cover" the modes of all answer entity embeddings [34].

**Training Objective:** Our objective is to minimize the distance between the Beta embedding of a query and its answers while maximizing the distance between the Beta embedding of the query and other random entities via negative sampling [6, 10], which we define as follows:

$$L = -\log \sigma \left( \gamma - \texttt{Dist}(v; q) \right) - \sum_{j=1}^{k} \frac{1}{k} \log \sigma \left( \texttt{Dist}(v_j'; q) - \gamma \right), \tag{6}$$

where $v \in [\![q]\!]$ belongs to the answer set of $q$, $v_j' \notin [\![q]\!]$ represents a random negative sample, and $\gamma$ denotes the margin. In the loss function, we use $k$ random negative samples and optimize the average.

**Discussion on Modeling Union:** With the De Morgan's laws (abbreviated as DM), we can naturally model union operation $S_1 \cup S_2$ with $\overline{\overline{S_1} \cap \overline{S_2}}$, which we can derive as a Beta embedding. However, according to the Theorem 1 in [10], in order to model any queries with the union operation, we must have a parameter dimensionality of $\Theta(M)$, where $M$ is of the same order as the number of entities [10]. The reason is that we need to model in the embedding space any subset of the entities. Q2B [10] overcomes this limitation by transforming queries into a disjunctive normal form (DNF) and only deals with union at the last step. Our DM modeling of union is also limited in this respect since the Beta embedding can be at most bi-modal and as a result, there are some union-based queries that BETAE cannot model in theory. However, in practice, union-based queries are constrained and we do not need to model all theoretically possible entity subsets. For example, a query "*List the union of European countries and tropical fruits.*" does not make sense; and we further learn high-dimensional Beta embeddings to alleviate the problem. Furthermore, our DM modeling is always linear w.r.t the number of union operations, while the DNF modeling is exponential in the worst case (with detailed discussion in Appendix B). Last but not least, BETAE can safely incorporate both the DNF modeling and DM modeling, and we show in the experiments that the two approaches work equally well in answering real-world queries.

**Inference:** Given a query $q$, BETAE directly embeds it as $\mathbf{q}$ by following the computation graph without the need to model intermediate entities. To obtain the final answer entities, we rank all the entities based on the distance defined in Eq. 5 in constant time using Locality Sensitive Hashing [13].

## 5 Experiments

In this section, we evaluate BETAE on multi-hop reasoning over standard KG benchmark datasets. Our experiments demonstrate that: (1) BETAE effectively answers arbitrary FOL queries. (2) BETAE outperforms less general methods [9, 10] on EPFO queries (containing only $\exists$, $\wedge$ and $\vee$) that these methods can handle. (3) The probabilistic embedding of a query corresponds well to its uncertainty.

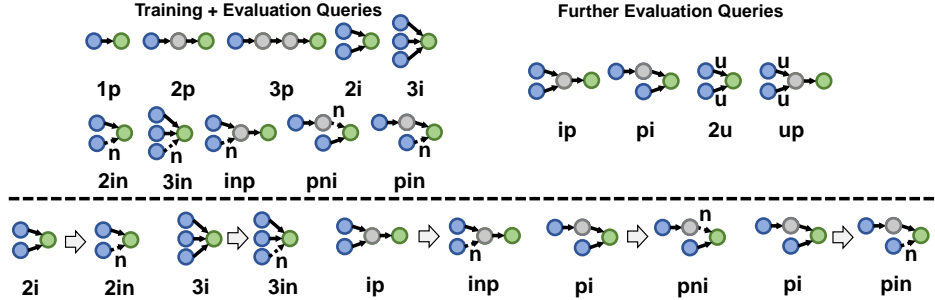

Figure 3: Top: Training and evaluation queries represented with their graphical structures, an abbreviation of the computation graph. Naming convention: $p$ projection, $i$ intersection, $n$ negation, $u$ union. Bottom: Query structures with negation used in our experiments.

| Dataset | Model | 1p | 2p | 3p | 2i | 3i | pi | ip | 2u DNF | 2u DM | up DNF | up DM | avg |
|---------|-------|-----|------|------|------|------|------|------|------|------|------|------|------|
| FB15k | BETAE | 65.1 | **25.7** | **24.7** | **55.8** | 66.5 | **43.9** | **28.1** | **40.1** | 25.0 | **25.2** | 25.4 | **41.6** |
| | Q2B | **68.0** | 21.0 | 14.2 | 55.1 | **66.5** | 39.4 | 26.1 | 35.1 | - | 16.7 | - | 38.0 |
| | GQE | 54.6 | 15.3 | 10.8 | 39.7 | 51.4 | 27.6 | 19.1 | 22.1 | - | 11.6 | - | 28.0 |
| FB15k-237 | BETAE | 39.0 | **10.9** | **10.0** | 28.8 | **42.5** | **22.4** | 12.6 | **12.4** | 11.1 | **9.7** | 9.9 | **20.9** |
| | Q2B | **40.6** | 9.4 | 6.8 | **29.5** | 42.3 | 21.2 | **12.6** | 11.3 | - | 7.6 | - | 20.1 |
| | GQE | 35.0 | 7.2 | 5.3 | 23.3 | 34.6 | 16.5 | 10.7 | 8.2 | - | 5.7 | - | 16.3 |
| NELL995 | BETAE | **53.0** | 13.0 | **11.4** | 37.6 | 47.5 | **24.1** | 14.3 | **12.2** | 11.0 | 8.5 | 8.6 | **24.6** |
| | Q2B | 42.2 | **14.0** | 11.2 | 33.3 | 44.5 | 22.4 | **16.8** | 11.3 | - | **10.3** | - | 22.9 |
| | GQE | 32.8 | 11.9 | 9.6 | 27.5 | 35.2 | 18.4 | 14.4 | 8.5 | - | 8.8 | - | 18.6 |

Table 1: MRR results (%) of BETAE, Q2B and GQE on answering EPFO ($\exists$, $\wedge$, $\vee$) queries.

## 5.1 Experiment Setup

Our experimental setup is focused on incomplete KGs and thus we measure performance only over answer entities that require (implicitly) imputing at least one edge. More precisely, given an incomplete KG, our goal is to obtain *non-trivial* answers to arbitrary FOL queries that *cannot* be discovered by directly traversing the KG. We use three standard KGs with official training/validation/test edge splits, FB15k [4], FB15k-237 [35] and NELL995 [27] and follow [10] for the preprocessing.

**Evaluation Protocol:** We follow the evaluation protocol in [10]. We first build three KGs: training KG $\mathcal{G}_{\texttt{train}}$, validation KG $\mathcal{G}_{\texttt{valid}}$, test KG $\mathcal{G}_{\texttt{test}}$ using training edges, training+validation edges, training+validation+test edges, respectively. Our evaluation focuses on incomplete KGs, so given a test (validation) query $q$, we are interested in discovering *non-trivial* answers $[\![q]\!]_{\texttt{test}} \backslash [\![q]\!]_{\texttt{val}}$ ($[\![q]\!]_{\texttt{val}} \backslash [\![q]\!]_{\texttt{train}}$). That is, answer entities where at least one edge needs to be imputed in order to create an answer path to that entity. For each non-trivial answer $v$ of a test query $q$, we rank it against non-answer entities $\mathcal{V} \backslash [\![q]\!]_{\texttt{test}}$. We denote the rank as $r$ and calculate the Mean Reciprocal Rank (MRR): $\frac{1}{r}$; and, Hits at $K$ (H@K): $\mathbf{1}[r \leq K]$ as evaluation metrics.

**Queries:** We base our queries on the 9 query structures proposed in Query2Box (Q2B) [10] and make two additional improvements. First, we notice that some test queries may have more than 5,000 answers. To make the task more challenging, we thus regenerate the same number of validation/test queries for each of the 9 structures, keeping only those with answers smaller than a threshold. We list the statistics of the new set of queries in Table 6 (in Appendix C). We evaluate BETAE on both the queries in Q2B and our new realistic queries, which are more challenging since they use the same training queries without any enforcement on the maximum number of answers for a fair comparison. Second, from the 9 structures we derive 5 new query structures with negation. As shown in Fig. 3, in order to create realistic structures with negation, we look at the 4 query structures with intersection ($2i/3i/ip/pi$) and perturb one edge to perform set complement before taking the intersection, resulting in $2in/3in/inp/pni/pin$ structures. Additional information about query generation is given in Appendix C.

As summarized in Fig. 3, our training and evaluation queries consist of the 5 conjunctive structures ($1p/2p/3p/2i/3i$) and also 5 novel structures with negation ($2in/3in/inp/pni/pin$). Furthermore, we also evaluate model's generalization ability which means answering queries with logical structures that the model has never seen during training. We further include $ip/pi/2u/up$ for evaluation.

| Dataset | Metrics | 2in | 3in | inp | pin | pni | avg |
|---------|---------|-----|-----|-----|-----|-----|-----|
| FB15k | MRR | 14.3 | 14.7 | 11.5 | 6.5 | 12.4 | 11.8 |
| | H@10 | 30.8 | 31.9 | 23.4 | 14.3 | 26.3 | 25.3 |
| FB15k-237 | MRR | 5.1 | 7.9 | 7.4 | 3.6 | 3.4 | 5.4 |
| | H@10 | 11.3 | 17.3 | 16.0 | 8.1 | 7.0 | 11.9 |
| NELL995 | MRR | 5.1 | 7.8 | 10.0 | 3.1 | 3.5 | 5.9 |
| | H@10 | 11.6 | 18.2 | 20.8 | 6.9 | 7.2 | 12.9 |

Table 2: MRR and H@10 results (%) of BETAE on answering queries with negation.

**Baselines:** We consider two state-of-the-art baselines for answering complex logical queries on KGs: Q2B [10] and GQE [9]. GQE embeds both queries and entities as point vectors in the Euclidean space; Q2B embeds the queries as hyper-rectangles (boxes) and entities as point vectors so that answers will be enclosed in the query box. Both methods design their corresponding projection and intersection operators, however, neither can handle the negation operation since the complement of a point/box in the Euclidean space is no longer a point/box. For fair comparison, we assign the same dimensionality to the embeddings of the three methods[2]. Note that since the baselines cannot model negation operation, the training set for the baselines only contain queries of the 5 conjunctive structures. We ran each method for 3 different random seeds after finetuning the hyperparameters. We list the hyperparameters, architectures and more details in Appendix D.

## 5.2 Modeling Arbitrary FOL Queries

**Modeling EPFO (containing only $\exists$, $\wedge$ and $\vee$) Queries:** First we compare BETAE with baselines that can only model queries with conjunction and disjunction (but no negation). Table 1 shows the MRR of the three methods. BETAE achieves on average 9.4%, 5.0% and 7.4% relative improvement MRR over previous state-of-the-art Q2B on FB15k, FB15k-237 and NELL995, respectively. We refer the reader to Tables 9 and 10 in Appendix E for the H@1 results. Again, on EPFO queries BETAE achieves better performance than the two baselines on all three datasets.

**DNF vs. DM:** As discussed in Sec. 4.3, we can model queries with disjunction in two ways: (1) transform them into disjunctive normal form (DNF); (2) represent disjunction with conjunction and negation using the De Morgan's laws (DM). We evaluate both modeling schemes (Table 1(right)). DNF modeling achieves slightly better results than DM since it is able to better represent disjunction with multi-modal embeddings. However, it also demonstrates that our DM modeling provides a nice approximation to the disjunction operation, and generalizes really well since the model is not trained on $2u$ and $up$ queries. Note that BETAE is very flexible and can use and improve both modeling approaches while the baselines can only use DNF since they cannot model the negation operation.

**Modeling Queries with Negation:** Next, we evaluate our model's ability to model queries with negation. We report both the MRR and H@10 results in Table 2. Note that answering queries with negation is challenging since only a small fraction of the training queries contain negation. As shown in Table 7 (Appendix), during training, the number of $2in/3in/inp/pin/pni$ queries is 10 times smaller than the number of conjunctive queries. Overall, BETAE generalizes well and provides the first embedding-based method that can handle arbitrary FOL queries.

## 5.3 Modeling the Uncertainty of Queries

We also investigate whether our Beta embeddings are able to capture uncertainty. The uncertainty of a (fuzzy) set can be characterized by its cardinality. Given a query with answer set $[\![q]\!]$, we aim to calculate the correlation between the differential entropy of the Beta embedding $p_{[\![\mathbf{q}]\!]}$ and the cardinality of the answer set $|[\![q]\!]|$. For comparison, Q2B embeds each query as a box, which can also model the uncertainty of the query by expanding/shrinking the box size. We consider two types of statistical correlations: Spearman's rank correlation coefficient (SRCC), which measures the statistical dependence between the rankings of two variables; and Pearson's correlation coefficient (PCC), which measures the linear correlation of the two variables. Table 3 and Table 11 (in Appendix E) show that BETAE achieves up to 77% better correlation than Q2B. We conclude that BETAE with Beta embeddings is able to capture query uncertainty. Furthermore, note that BETAE naturally learns this property without any regularization to impose the correlation during training.

| Dataset | Model | 1p | 2p | 3p | 2i | 3i | pi | ip | 2in | 3in | inp | pin | pni |
|---|---|---|---|---|---|---|---|---|---|---|---|---|---|
| FB15k | Q2B | 0.301 | 0.219 | 0.262 | 0.331 | 0.270 | 0.297 | 0.139 | - | - | - | - | - |
| | BETAE | **0.373** | **0.478** | **0.472** | **0.572** | **0.397** | **0.519** | **0.421** | 0.622 | 0.548 | 0.459 | 0.465 | 0.608 |
| FB15k-237 | Q2B | 0.184 | 0.226 | 0.269 | 0.347 | 0.436 | 0.361 | 0.199 | - | - | - | - | - |
| | BETAE | **0.396** | **0.503** | **0.569** | **0.598** | **0.516** | **0.540** | **0.439** | 0.685 | 0.579 | 0.511 | 0.468 | 0.671 |
| NELL995 | Q2B | 0.154 | 0.288 | 0.305 | 0.380 | 0.410 | 0.361 | 0.345 | - | - | - | - | - |
| | BETAE | **0.423** | **0.552** | **0.564** | **0.594** | **0.610** | **0.598** | **0.535** | 0.711 | 0.595 | 0.354 | 0.447 | 0.639 |

Table 3: Spearman's rank correlation between learned embedding (differential entropy for BETAE, box size for Q2B) and the number of answers of queries. BETAE shows up to 77% relative improvement.

| 1p | 2p | 3p | 2i | 3i | pi | ip | 2in | 3in | inp | pin | pni |
|---|---|---|---|---|---|---|---|---|---|---|---|
| 0.825 | 0.766 | 0.793 | 0.909 | 0.933 | 0.868 | 0.798 | 0.865 | 0.93 | 0.801 | 0.809 | 0.848 |

Table 4: ROC-AUC score of BETAE for all the 12 query structures on classification of queries with/without answers on the NELL dataset.

**Modeling Queries without Answers:** Since BETAE can effectively model the **uncertainty** of a given query, we can use the differential entropy of the query embedding as a measure to represent whether the query is an empty set (has no answers). For evaluation, we randomly generated 4k queries without answers and 4k queries with more than 5 answers for each of the 12 query structures on NELL. Then we calculate the differential entropy of the embeddings of each query with a trained BETAE and use this to classify whether a query has answers. As a result, we find an ROC-AUC score of 0.844 and list the ROC-AUC score of each query structure in Table 4. These results suggest that BETAE can naturally model queries without answers, since (1) we did not explicitly train BETAE to optimize for correlation between the differential entropy and the cardinality of the answer set; (2) we did not train BETAE on queries with empty answers.

# 6 Conclusion

We have presented BETAE, the first embedding-based method that could handle arbitrary FOL queries on KGs. Given a query, BETAE embeds it into Beta distributions using probabilistic logical operators by following the computation graph in a scalable manner. Extensive experimental results show that BETAE significantly outperforms previous state-of-the-art, which can only handle a subset of FOL, in answering arbitrary logical queries as well as modeling the uncertainty.

# Broader Impact

BETAE gives rise to the first method that handles all logical operators in large heterogeneous KGs. It will greatly increase the scalability and capability of multi-hop reasoning over real-world KGs and heterogenous networks.

One potential risk is that the model may make undesirable predictions in a completely random KG, or a KG manipulated by adversarial and malicious attacks [36, 37]. Recent progress on adversarial attacks [36, 37] have shown that manipulation of the KG structure may effectively deteriorate the performance of embedding-based methods. And this may mislead the users and cause negative impact. We will continue to work on this direction to design more robust KG embeddings. Alternatively, this issue can also be alleviated through human regularization of real-world KGs.

# Acknowledgments and Disclosure of Funding

We thank Shengjia Zhao, Rex Ying, Jiaxuan You, Weihua Hu, Tailin Wu and Pan Li for discussions, and Rok Sosic for providing feedback on our manuscript. Hongyu Ren is supported by the Masason Foundation Fellowship. Jure Leskovec is a Chan Zuckerberg Biohub investigator. We also gratefully acknowledge the support of DARPA under Nos. FA865018C7880 (ASED), N660011924033 (MCS); ARO under Nos. W911NF-16-1-0342 (MURI), W911NF-16-1-0171 (DURIP); NSF under Nos. OAC-1835598 (CINES), OAC-1934578 (HDR), CCF-1918940 (Expeditions), IIS-2030477 (RAPID); Stanford Data Science Initiative, Wu Tsai Neurosciences Institute, Chan Zuckerberg Biohub, Amazon, Boeing, JPMorgan Chase, Docomo, Hitachi, JD.com, KDDI, NVIDIA, Dell.

## Footnotes

[1]Note that we do not consider FOL queries with universal quantification ($\forall$) in this paper. Queries with universal quantification do not apply in real-world KGs since no entity connects with all the other entities.

[2]If GQE has embeddings of dimension $2n$, then Q2B has embeddings of $n$ since it needs to model both the center and offset of a box, and BETAE also has $n$ beta distributions since each has two parameters, $\alpha$ and $\beta$.

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
