[Supplementary Material]

# Appendix

## A  Proof for Proposition 1

We restate the proposition 1 and its proof here.

**Proposition 2.** *Given the probabilistic logical operators $\mathcal{I}$ and $\mathcal{N}$ defined in Sec. 4.2,* BETAE *has the following properties:*

1. *Given Beta embedding* $\mathbf{S}$, $\mathbf{S}$ *is a fixed point of* $\mathcal{N} \circ \mathcal{N}$: $\mathcal{N}(\mathcal{N}(\mathbf{S})) = \mathbf{S}$.

2. *Given Beta embedding* $\mathbf{S}$, *we have* $\mathcal{I}(\{\mathbf{S}, \mathbf{S}, \dots, \mathbf{S}\}) = \mathbf{S}$.

*Proof.* For the first property, the probabilistic negation operator $\mathcal{N}$ takes the reciprocal of the parameters of the input Beta embeddings. If we apply $\mathcal{N}$ twice, it naturally equals the input Beta embeddings. For the second property, the probabilistic intersection operator $\mathcal{I}$ takes the weighted product of the PDFs of the input Beta embeddings, and according to Eq. 4, the parameters of the output Beta embeddings are linear interpolation of the parameters of the input Beta embeddings. Then we naturally have $\mathbf{S} = \mathcal{I}(\{\mathbf{S}, \dots, \mathbf{S}\})$. □

## B  Computation Complexity of DM and DNF

Here we discuss the computation complexity of representing any given FOL query using the De Morgan's laws (DM) and the disjunctive normal form (DNF). Given a FOL query $q$, representing $q$ with DNF may in the worst case creates exponential number of atomic formulas. For example, transforming a valid FOL query $(q_{11} \lor q_{12}) \land (q_{21} \lor q_{22}) \cdots \land (q_{n1} \lor q_{n2})$ leads to exponential explosion, resulting in a query with $2^n$ number of formulas in the DNF. For DM, since we could always represent a disjunction operation with three negation operation and one conjunction operation: $q_1 \lor q_2 = \neg(\neg q_1 \land \neg q_2)$, which is a constant. Hence, the DM modeling only scales linearly.

## C  Query Generation and Statistics

**Generation of EPFO (with $\exists$, $\lor$ and $\land$) Queries:** Following [10], we generate the 9 EPFO query structures in a similar manner. Given the three KGs, and its training/validation/test edge splits, which is shown in Table 5, we first create $\mathcal{G}_{\text{train}}$, $\mathcal{G}_{\text{valid}}$, $\mathcal{G}_{\text{test}}$ as discussed in Sec. 5.1. Then for each query structure, we use pre-order traversal starting from the target node/answer to assign an entity/relation to each node/edge iteratively until we instantiate every anchor nodes (the root of the query structure). After the instantiation of a query, we could perform post-order traversal to achieve the answers of this query. And for validation/test queries, we explicitly filter out ones that do not exist non-trivial answers, *i.e.*, they can be fully answered in $\mathcal{G}_{\text{train}}/\mathcal{G}_{\text{valid}}$. Different from the dataset in [10], where the maximum number of test queries may exceed 5,000, we set a bar for the number of answers one query has, and additionally filter out unrealistic queries with more than 100 answers. We list the average number of answers the new test queries have in Table 6 and the number of training/validation/test queries in Table 7.

| Dataset | Entities | Relations | Training Edges | Validation Edges | Test Edges | Total Edges |
|---|---|---|---|---|---|---|
| FB15k | 14,951 | 1,345 | 483,142 | 50,000 | 59,071 | 592,213 |
| FB15k-237 | 14,505 | 237 | 272,115 | 17,526 | 20,438 | 310,079 |
| NELL995 | 63,361 | 200 | 114,213 | 14,324 | 14,267 | 142,804 |

Table 5: Knowledge graph dataset statistics as well as training, validation and test edge splits.

| Dataset | 1p | 2p | 3p | 2i | 3i | ip | pi | 2u | up | 2in | 3in | inp | pin | pni |
|---|---|---|---|---|---|---|---|---|---|---|---|---|---|---|
| FB15k | 1.7 | 19.6 | 24.4 | 8.0 | 5.2 | 18.3 | 12.5 | 18.9 | 23.8 | 15.9 | 14.6 | 19.8 | 21.6 | 16.9 |
| FB15k-237 | 1.7 | 17.3 | 24.3 | 6.9 | 4.5 | 17.7 | 10.4 | 19.6 | 24.3 | 16.3 | 13.4 | 19.5 | 21.7 | 18.2 |
| NELL995 | 1.6 | 14.9 | 17.5 | 5.7 | 6.0 | 17.4 | 11.9 | 14.9 | 19.0 | 12.9 | 11.1 | 12.9 | 16.0 | 13.0 |

Table 6: Average number of answers of test queries in our new dataset.

| Queries | Training | | Validation | | Test | |
|---------|----------|---|------------|---|------|---|
| Dataset | 1p/2p/3p/2i/3i | 2in/3in/inp/pin/pni | 1p | others | 1p | others |
| FB15k | 273,710 | 27,371 | 59,097 | 8,000 | 67,016 | 8,000 |
| FB15k-237 | 149,689 | 14,968 | 20,101 | 5,000 | 22,812 | 5,000 |
| NELL995 | 107,982 | 10,798 | 16,927 | 4,000 | 17,034 | 4,000 |

Table 7: Number of training, validation, and test queries generated for different query structures.

**Generation of Queries with Negation:** For the additional queries with negation, we derive 5 new query structures from the 9 EPFO structures. Specifically, as shown in Fig. 3, we only consider query structures with intersection for the derivation of queries with negation. The reason is that queries with negation are only realistic if we take negation with an intersection together. Consider the following example, where negation is not taken with intersection, "*List all the entities on KG that is not European countries.*", then both "*apple*" and "*computer*" will be the answers. However, realistic queries will be like "*List all the countries on KG that is not European countries.*", which requires an intersection operation. In this regard, We modify one edge of the intersection to further incorporate negation, thus we derive $2in$ from $2i$, $3in$ from $3i$, $inp$ from $ip$, $pin$ and $pni$ from $pi$. Note that following the 9 EPFO structures, we also enforce that all queries with negation have at most 100 answers.

## D   Experimental Details

We implement our code using Pytorch. We use the implementation of the two baselines GQE [9] and Q2B [10] in https://github.com/hyren/query2box. We finetune the hyperparameters for the three methods including number of embedding dimensions from $\{200, 400, 800\}$ and the learning rate from $\{1e^{-4}, 5e^{-3}, 1e^{-3}\}$, batch size from $\{128, 256, 512\}$, and the negative sample size from $\{32, 64, 128\}$, the margin $\gamma$ from $\{20, 30, 40, 50, 60, 70\}$. We list the hyperparameters of each model in the Table 8. Additionally, for our BETAE, we finetune the structure of the probabilistic projection operator MLP$_r$ and the attention module MLP$_{\text{Att}}$. For both modules, we implement a three-layer MLP with 512 latent dimension and ReLU activation.

| | embedding dim | learning rate | batch size | negative sample size | margin |
|---|---|---|---|---|---|
| GQE | 800 | 0.0005 | 512 | 128 | 30 |
| Q2B | 400 | 0.0005 | 512 | 128 | 30 |
| BETAE | 400 | 0.0005 | 512 | 128 | 60 |

Table 8: Hyperparameters used for each method.

Each single experiment is run on a single NVIDIA GeForce RTX 2080 TI GPU, and we run each method for 300k iterations.

## E   Additional Experimental Results

Here we list some additional experimental results.

We show in Table 1 the MRR results of the three methods on answering EPFO queries. Our methods show a significant improvement over the two baselines in all three datasets.

We show in Table 10 the MRR results of the three methods on answering EPFO queries in the dataset proposed in [10], where the queries may have more than 5,000 answers. Our method is still better than the two baselines.

We show in Table 11 the Pearson correlation coefficient between the learned embedding and the number of answers of queries. Our method is better than the baseline Q2B in measuring the uncertainty of the queries.

| Dataset | Model | 1p | 2p | 3p | 2i | 3i | pi | ip | 2u | | up | | avg |
|---|---|---|---|---|---|---|---|---|---|---|---|---|---|
| | | | | | | | | | DNF | DM | DNF | DM | |
| FB15k | BETAE | **52.0** | **17.0** | **16.9** | **43.5** | **55.3** | **32.3** | **19.3** | **28.1** | 17.0 | **16.9** | 17.4 | **31.3** |
| | Q2B | **52.0** | 12.7 | 7.8 | 40.5 | 53.4 | 26.7 | 16.7 | 22.0 | - | 9.4 | - | 26.8 |
| | GQE | 34.2 | 8.3 | 5.0 | 23.8 | 34.9 | 15.5 | 11.2 | 11.5 | - | 5.6 | - | 16.6 |
| FB15k-237 | BETAE | **28.9** | **5.5** | **4.9** | **18.3** | **31.7** | **14.0** | 6.7 | **6.3** | 6.1 | **4.6** | 4.8 | **13.4** |
| | Q2B | 28.3 | 4.1 | 3.0 | 17.5 | 29.5 | 12.3 | **7.1** | 5.2 | - | 3.3 | - | 12.3 |
| | GQE | 22.4 | 2.8 | 2.1 | 11.7 | 20.9 | 8.4 | 5.7 | 3.3 | - | 2.1 | - | 8.8 |
| NELL995 | BETAE | **43.5** | 8.1 | **7.0** | **27.2** | **36.5** | **17.4** | 9.3 | **6.9** | 6.0 | 4.7 | 4.7 | **17.8** |
| | Q2B | 23.8 | **8.7** | 6.9 | 20.3 | 31.5 | 14.3 | **10.7** | 5.0 | - | **6.0** | - | 14.1 |
| | GQE | 15.4 | 6.7 | 5.0 | 14.3 | 20.4 | 10.6 | 9.0 | 2.9 | - | 5.0 | - | 9.9 |

Table 9: H@1 results (%) of BETAE, Q2B and GQE on answering EPFO ($\exists$, $\land$, $\lor$) queries.

| Dataset | Model | 1p | 2p | 3p | 2i | 3i | pi | ip | 2u | up | avg |
|---|---|---|---|---|---|---|---|---|---|---|---|
| FB15k | BETAE | 65.0 | **42.1** | **37.8** | **52.9** | **64.0** | **41.5** | **22.9** | 48.8 | 26.9 | **44.6** |
| | Q2B | **67.1** | 38.0 | 27.5 | 49.2 | 62.8 | 36.2 | 19.2 | **49.0** | **28.9** | 42.0 |
| | GQE | 54.6 | 30.5 | 22.2 | 37.7 | 48.4 | 24.8 | 14.7 | 33.8 | 24.7 | 32.4 |
| FB15k-237 | BETAE | 39.1 | **24.2** | **20.4** | **28.1** | **39.2** | **19.4** | **10.6** | **22.0** | 17.0 | **24.4** |
| | Q2B | **40.3** | 22.8 | 17.5 | 27.5 | 37.9 | 18.5 | 10.5 | 20.5 | **17.4** | 23.6 |
| | GQE | 35.0 | 19.0 | 14.4 | 22.0 | 31.2 | 14.6 | 8.8 | 15.0 | 14.6 | 19.4 |
| NELL995 | BETAE | **53.0** | **27.5** | **28.1** | **32.9** | **45.1** | **21.8** | 10.4 | **38.6** | **19.6** | **30.7** |
| | Q2B | 41.8 | 22.9 | 20.8 | 28.6 | 41.2 | 19.9 | **12.3** | 26.9 | 15.5 | 25.5 |
| | GQE | 32.8 | 19.3 | 17.9 | 23.1 | 31.9 | 16.2 | 10.3 | 17.3 | 13.1 | 20.2 |

Table 10: MRR results (%) on queries from [10], where we show that we are also able to achieve higher performance than baselines Q2B and GQE on all three KGs.

| Dataset | Model | 1p | 2p | 3p | 2i | 3i | pi | ip | 2in | 3in | inp | pin | pni |
|---|---|---|---|---|---|---|---|---|---|---|---|---|---|
| FB15k | Q2B | 0.075 | 0.217 | 0.258 | 0.285 | 0.226 | 0.245 | 0.133 | - | - | - | - | - |
| | BETAE | **0.216** | **0.357** | **0.383** | **0.386** | **0.299** | **0.311** | **0.312** | 0.438 | 0.413 | 0.343 | 0.360 | 0.442 |
| FB15k-237 | Q2B | 0.017 | 0.194 | 0.261 | **0.366** | **0.488** | **0.335** | 0.197 | - | - | - | - | - |
| | BETAE | **0.225** | **0.365** | **0.450** | 0.362 | 0.307 | 0.319 | **0.332** | 0.464 | 0.409 | 0.390 | 0.361 | 0.484 |
| NELL995 | Q2B | 0.068 | 0.211 | 0.306 | 0.362 | 0.287 | 0.240 | 0.338 | - | - | - | - | - |
| | BETAE | **0.236** | **0.403** | **0.433** | **0.404** | **0.385** | **0.403** | **0.403** | 0.515 | 0.514 | 0.255 | 0.354 | 0.455 |

Table 11: Pearson correlation coefficient between learned embedding (differential entropy for BETAE, box size for Q2B) and the number of answers of queries (grouped by different query type). Ours achieve higher correlation coefficient.