[Reviews · NeurIPS 2020]

Review 1

Summary and Contributions: This paper introduces a method to handle queries in first order logic on knowledge graphs. Each entity is represented by a vector of Beta distributions and relationships are modelled as neural networks that operate on beta embeddings. Previous work has modelled entity embeddings as single points and boxes, neither of which support negation. The authors explain how the space of beta embeddings admit a natural negation operation which inverts the probability mass represented by the distribution. This enables their approach to answer arbitrary first-order logic queries, where previous approaches were limited to queries involving the logical and, or and existential operators.

Strengths: The approach described here is novel and delivers both better performance and greater expressivity over previous work. The problem of answering logical queries on a knowledge graph is an interesting one and this work is likely to be of wider interest to the parts of the NeurIPS community that are interested in knowledge graph inference and related problems. The experiments are well-chosen and demonstrate the key claims of the paper: - Their approach improves over the baselines on the set of queries that all approaches can answer - Their approach gives good performance on queries involving negation - Their alternative approach gives a comparable performance to DNF, but with improved computational complexity

Weaknesses: In Table 1, H@1 is used but in Table 2 H@10 and MRR are used in Table 2. Why not use the same metric(s) in both tables? -- The authors have addressed this in their response.

Correctness: The empirical methodology seems correct to me and the results of additional experiments provided in the appendix are useful to better understand model performance.

Clarity: The paper is was well-written and easy to follow.

Relation to Prior Work: The most relevant prior work is discussed and included as baselines in the experiments. The difference in terms of architecture and improvement is presented clearly.

Reproducibility: Yes

Additional Feedback:


Review 2

Summary and Contributions: The paper proposes a probabilistic embedding framework for answering arbitrary first-order logic queries over knowledge graphs where relations and entities are represented by the Beta distribution, and logical operations are performed in the vector space over the probabilistic embeddings. It is the first approach that can apply arbitrary first-order logic in the embeddings space.

Strengths: This paper seems to be a useful contribution to the literature on answering first-order logic queries on knowledge graphs. The author proposed a novel probabilistic negation operator in the embedding space to deal with logical negation, which has not been solved in literature. Besides, by correlating differential entropy of the Beta embedding with the uncertainty of the queries, the proposed model can not only operate all first-order logic operations but also captures the uncertainty. The review thinks that this paper would be interesting to the knowledge graph community.

Weaknesses: 1. The theoretical analysis of the model is insufficient. For example, the author does not give an analysis of the full expressiveness of the model. That is, given any world with correct answers of some first-order logic queries W and false answers Wc, does there exist an assignment for model parameters that correctly classifies the entities in W and Wc? The reviewer is especially curious about the theoretical analysis of the proposed probabilistic negation operator because there are no comparative empirical results on answering queries with negation (all existing models cannot deal with negation). 2. The empirical evaluation is inadequate. On EPFO queries, the author compared the proposed model only with two baselines. It would be nicer if the authors can adapt the FOL queries to the settings of other multi-hop reasoning models such as [1] and [2] and compare the proposed model with these path-based methods. 3. The author did not sufficiently discuss the experimental results. For instance, why are the improvements on NELL995 considerably larger than FB15k? Especially for 1p queries given in Table 1, both Q2B and GQE achieved worse results on NELL995 (23.4/15.4) than the results on FB15k-237 (28.0/22.4) while BETAE achieved a much better result on NELL995 (42.8 on NELL995 28.0 on FB15k-237 ). What would be the reason for that? [1] S. Guo, Q. Wang, L. Wang, B. Wang, and L. Guo, "Knowledge graph embedding with iterative guidance from soft rules," in AAAI Conference on Artificial Intelligence (AAAI), 2018. [2] R. Das, A. Neelakantan, D. Belanger, and A. McCallum, "Chains of reasoning over entities, relations, and text using recurrent neural networks," in European Chapter of the Association for Computational Linguistics (EACL), pp. 132–141, 2017.

Correctness: The technical content and the empirical methodology of the paper appears to be correct. But the paper does not clearly claim whether the results are raw or filtered. For the definition of filtered results, please refer to [3]. [3] A. Bordes, N. Usunier, A. Garcia-Duran, J.Weston, and O. Yakhnenko, "Translating embeddings for modeling multi-relational data," in Advances in Neural Information Processing Systems (NeurIPS), pp. 2787–2795, 2013.

Clarity: Generally, the paper is well written and structured clearly, albeit some tiny things to correct. First, the uncertainty seems to be a highlighting point of the model, but the claim is not well supported. The author should give more details about the correlation between the differential entropy of the Beta embedding with the uncertainty of the queries in Section 5.3. Additionally, the paper should clearly claim in the introduction and the experimental part that the model starts from a well-structured firstorder logic query but not from a natural language question. For example, in line 29 "A concrete example of the computation graph for the query 'List the presidents of European countries that have never held the World Cup' is shown in Fig. 1.", it seems like the proposed model can directly answer the natural language query where they can convert this query into a first-order logic form. Readers might be confused here.

Relation to Prior Work: Yes. The paper discussed existing knowledge graph embedding methods that can capture uncertainties of entities and relations on KGs. The major difference here is that the proposed method learns probabilistic embeddings for complex queries and designs a set of neural logical operators in the embedding space to perform multi-hop reasoning over knowledge graphs.

Reproducibility: Yes

Additional Feedback: BetaE uses De Morgan's law to apply disjunctions. The disjunctions of several beta embeddings should be multi-modal. If there are more than two peaks, can BetaE have a multi-modal distribution in each single dimension?


Review 3

Summary and Contributions: This paper presents a method of representing entities and sets as a fixed number of independent Beta distributions. While there is no distinct notion of set membership, queries can be represented in the same manner, and answer suitability is scored using the KL divergence between the query and answer representations. The paper presents operations analagous to negation and intersection, that operate on the chosen representatino. It alsos present approximation to union that applies De-Morgan's law to the negation and intersection operations. This is just an approximation, the Beta distribution cannot properly model multiple modes, and authors point out that there is no proper way to represent union without an exploding parameter space. The paper presents an evaluation of the model's ability to model compositional queries with a variety of structures, some of which are unseen at training time. Comparison is made to other recent work on modeling compositional queries and the Beta embeddings perform better at better at ranking entity answers than a related box-embedding approach. There is also a brief discussion of how the Beta distribution can capture the cardinality of the answer set naturally.

Strengths: - Novel contribution. Good addition family of approaches that model entities and sets as points, volumes, Gaussian distributions. - Present reasonable analogues of intersection and negation. Negation not possible in volume-based approaches like box embeddings. - Also present approximation to union that is limited, but beyond previous apporaches. - Very relevant to the NeurIPS community.

Weaknesses: - No crisp concept of containment, unlike box embeddings. - Instead rely on a KL divergence distance metric, and evaluate this purely on its ability to rank results to queries that have non-empty result-sets. - Not clear that this distance metric is calibrated, or that it could be used to implement an existential operator which resolves to true or false. - Some limitations of the union operation are discussed but I would like to see some empirical evaluation of exactly what this operation can or cannot do. While the application of De-Morgan's law is interesting, I believe it is not a-priori valid when applied to the negation and intersection operations presented here (which are not exact counterparts of logical negation and intersection). Also it is clear that the Beta distribution cannot represent the multimodal distributions that I would expect as a result of many union operations. - Comparison is made to other efforts to model compositional queries. Since the representation itself is new, I would also like to see comparison to the much larger literature from modeling individual triples (presented here as queries of type 1p).

Correctness: I believe that the claims are correct. However, the paper could benefit from a discussion of how the various operators presented (negation, intersection, union) are *not* direct counterparts of their logical antecedants. Similarly, while there is some discussion of the correspondance between the differential entropy of an embedding and the cardinality of the set that it covers, I would like to see a distinct discussion of how this approach could be applied to non-ranking tasks. And what would an empty set look like?

Clarity: The paper is very well written and clear.

Relation to Prior Work: There is a good connection to previous work and this work is clearly novel. However, I feel that the work would benefit from direct empirical comparison to the much larger body of work on KG completion for single hop queries (1p queries in this work).

Reproducibility: Yes

Additional Feedback:

[Meta-Review · NeurIPS 2020]

The paper introduces a new method to query knowledge graphs via probabilistic embeddings based on the Beta distribution. The paper is well written and relevant to the NeurIPS community. All reviewers and the AC support acceptance of the paper for its contributions, notably since it proposes a novel and promising approach that enables logical negation and FOL queries on KG embeddings and as such extends the applicability of embeddings for KG inference tasks. However, please consider revising your paper to take feedback from reviewers into account e.g., in particular regarding the concerns raised related to empirical evaluation and theoretical analysis.